# Simplified Phenomenological Model for Ferroelectric Micro-Actuator [note 1]

**DOI:** 10.3390/mi14071355

**Published:** 2023-06-30

**Authors:** Binh Huy Nguyen, Guilherme Brondani Torri, Maja Zunic, Véronique Rochus

**Affiliations:** Sensor and Actuator Technology, imec, Kapeldreef 75, 3001 Leuven, Belgium

**Keywords:** phenomenological, ferroelectric, piezoelectric, PZT

## Abstract

As smart structures are becoming increasingly ubiquitous in our daily life, the need for efficient modeling electromechanical coupling devices is also rapidly advancing. Smart structures are often made of piezoelectric materials such as lead zirconate titanate (PZT), which exhibits strong nonlinear behavior known as hysteresis effect under a large applied electric field. There have been numerous modeling techniques that are able to capture such an effect; some techniques are suitable for obtaining physical insights into the micro-structure of the material, while other techniques are better-suited to practical structural analyses. In this paper, we aim to achieve the latter. We propose a simplified phenomenological macroscopic model of a nonlinear ferroelectric actuator. The assumption is based on the direct relation between the irreversible strain and irreversible electric field, and the consequently irreversible polarization. The proposed model is then implemented in a finite element framework, in which the main features such as local return mapping and the tangent moduli are derived. The outcomes of the model are compared and validated with experimental data. Therefore, the development presented in this paper can be a useful tool for the modeling of nonlinear ferroelectric actuators.

## 1. Introduction

In recent years, the pressing need to characterize the interactions between machines and the physical world has driven the high demand for the development of micro-electromechanical systems (MEMS). At the heart of MEMS’ design is the electromechanical materials that are capable of reciprocal converting between electrical and mechanical energy. An important subclass of electromechanical coupling material is piezoelectric materials, such as lead zirconate titanate (PZT). For applications that function at a low electric field, the piezoelectric material behaves linearly and its modeling is straightforward. However, in applications where a large electric field is required, the piezoelectric material exhibits nonlinear behavior with a hysteresis effect. This behavior can be categorized as a sub-class of piezoelectricity, namely ferroelectricity. In contrast to piezoelectric material, ferroelectric materials are characterized by a non-zero spontaneous polarization in the absence of an applied electric field, and the direction of this spontaneous polarization can be altered by changing the direction of the electric field. A detailed description of ferroelectricity can be found in other excellent review articles [1,2,3,4].

Microscopically, in PZT, switching polarization occurs when the titanium ions switch from one equilibrium position to another. The occurrence of such behavior in a single crystal has motivated the construction of the micro-electromechanical material model. In the first attempt to create this model, Hwang et al. [5] considered a ferroelectric polycrystal as a set of randomly oriented single crystals of a mono-domain without domain walls, such that the remanent state can be simplified to belong to six possible configurations. Under externally applied fields, the discrete transformation from one domain invariant to another is postulated via a critical energy-switching criterion. Subsequently, the response of the polycrystal was averaged based on the Reuss approximation of uniform stress and electric field. To account for interactions between grains, a modified energy criterion [6,7,8,9] or self-consistent homogenization techniques [10,11] were developed. Furthermore, to relax the assumption of complete switching that might induce instability, Huber et al. [10] proposed an incremental switching theory, in which the transformation from one state to another occurs continuously, followed by a self-consistent mean field homogenization to predict the overall response of polycrystal ferroelectric materials. Several finite-element implementations of a micro-electromechanical model were introduced [6,9,12,13,14,15] that could demonstrate their ability to capture the underlying physical switching. However, these might not be practical for structural analysis due to their high computational cost. There have only been a few attempts to achieve a more affordable micro-electromechanical material model. One notable work proposed replacing the random grain orientation distribution with a deterministic set of directions [16].

However, the phenomenological approach describes the ferroelectric response at a macroscopic level such that the computational cost is more favorable for structural analysis. In terms of numerical efficiency, the Preisach model seems to be the most convenient approach, in which the irreversible polarization can be reconstructed from the measured hysteresis polarization loop. In other words, a hysteresis operator that takes the applied electric field as the input and yields polarization as the output is determined by the fitting parameters from the hysteresis curve. The Preisach operator approach originated from ferromagnetic studies [17,18], which then naturally lend themselves to ferroelectricity [5,19], where irreversible polarization is obtained from the Preisach operator. Later, the irreversible strain can be directly related to the irreversible polarization or be reconstructed separately from the butterfly strain loop [20]. Once these operators are determined, they can be incorporated into an efficient finite-element framework [21,22]. The Preisach operator was also successfully incorporated into structural elements such as beams and shells in the works of Butz et al. [23] and Schulz et al. [24].

However, it should be noted that the Preisach model is not thermodynamically consistent. A thermodynamically consistent approach would strictly satisfy the second law of thermodynamics by introducing a dissipation potential. The dissipation potential and the free energy describe the ferroelectric material state via state variables. Specifically, the polarization switching phenomenon is characterized through a set of internal variables, such as remanent strain and remanant polarization. The first macroscopic phenomenological description of ferroelectricity was proposed by Chen et al. [25,26], who introduced the dynamic response of the electric displacement of a poled ferroelectric material. The theoretical foundation of a thermodynamic framework was laid by Bassiouny and Maugin [27,28,29]. In their framework, they establish an analogy to plasticity theory, in which the Helmholtz free energy is attributed from observable reversible variables and history-dependent internal variables. Strictly following the Clausius–Duhem inequality, the constitute laws and evolution form of the flux variables can be obtained for isotropic and kinematic hardening rules. Cock and McMeeking [30] later introduced an explicit expression for the yield function that determines when the switching process takes place with a demonstration of a 1D model. This framework was later rigorously refined by Landis [31] for a fully coupled, multi-axial in 3D model. In this work, closed-form tangent moduli were obtained and exhibited a symmetric form. Moreover, different switching surfaces were proposed and shown to agree with the experimental data. Such an accurate model comes at high cost, as nine internal variables were introduced and posed challenges to the determination coefficients for switching functions. Thus, a simplified version of Landis’s work was presented by McMeeking [32], in which a one-to-one relation between irreversible strain and polarization is assumed, such that the number of internal variables can be reduced. In a parallel body of works, Kamlah et al. [33], also prepared a macroscopic framework that is suitable for finite-element implementations. In their formulation, the irreversible strain is aligned with the direction of the applied electric field and directly related to the irreversible polarization. Additionally, four different loading criteria were introduced, two of which signify the irreversible processes, while the other two govern the fully poled saturated state. By refining the work of Kamlah et al. [1,33,34], with the additive decomposition of irreversible polarization stemming from the electric field and stress origin, Elhadrouz [35] also presented constitutive laws for ferroelectric and ferroelastic materials with the derivation of tangent moduli that could be suitable for finite-element implementations. A similar work has been proposed by Schwaab et al. [36] using the mixed finite-element framework.

A combination of a physically based microscopic model and phenomenologically based macroscopic model was also proposed. The first work in this direction was proposed by Zhou and Chattopadhyay [37], where the evolution of electric polarization with a varied electric field was motivated by the change in the dipole moment of the unit cell. Moreover, by using two sets of internal state variables, one of which is a texture tensor characterizing the orientation distribution function and a vector value for describing the macroscopic irreversible states, Mehling et al. [38] proposed a microscopically motivated phenomenological model, developing upon the theory framework of Kamlah. More recently, Stark et al. [39,40] introduced a hybrid framework, in which the volume fractions of the domain variants are chosen as the internal state variables.

To model a thin-film structure, a mesoscopic model, such as the phase-field method, is often a favorable choice [41]. The phase-field model also describes phenomenological behavior; however, the total polarization or spontaneous polarization is governed by a time-dependent diffusion equation [42,43,44,45]. Although phase-field method is a very powerful tool to describe the phase-transition effect, unfortunately it has a high computational cost and is not feasible for structural analysis. Moreover, material coefficients in the phase-field method are not trivially obtained from experiments but extracted from lower-scale modeling, such as density functional theory [46,47].

It is of interest to point out that, in the works of McMeeking and Landis [31,32], the saturated or locked-up phenomenon of remanent polarization or strain were introduced through the choice of hardening function, meaning that only one switching function is required. A similar framework was presented by Schroder [48,49,50]; however, the invariant of the state variables (reversible and irreversible) was chosen to be the argument of the energy and dissipation functions, together with a finite-element formulation. All the preceding works have utilized irreversible strain and polarization as the internal variables, which could entail an additional change in variables in a finite-element formulation where displacement and electric potential are considered as degrees of freedom. To circumvent this issue, instead of using the hybrid formulation from Ghandi [51] or performing two-step solving procedure from Kamlah et al. [34], Klinkel [52] proposed a fully coupled phenomenological model that employs irreversible strain and irreversible electric fields as internal variables. This work is as rigorous as the work from Landis [31], yet a detailed return-mapping and finite-element formulation were also provided. It should be emphasized that, in the modeling of ferroelectric devices, besides a reliable constitute model, numerical implementation is also of importance. For the computational aspects, readers are encouraged to read the works of Semenov [53] and Stark [54], which contain a detailed discussion on the return mapping and provide a derivation of the tangent moduli.

In this work, we present a simplified phenomenological model, which is formulated based on the irreversible electric field. The proposed model is inspired by the work of Klinkel [52]; however, we invoke the one-to-one relation between irreversible electric field and strain. By combining these two main features, the proposed model is not only advantageous for the numerical formulation and modeling but can also reduce the number of internal variables. Specifically, the evolution of the irreversible electric field can be determined from a simple dissipation potential that invokes only the coercive polarization. As a result, the determination of tangent elastic, piezoelectric and dielectric tensors, which are necessary for finite-element solutions, can be facilitated, as only the differentiation with respect to the irreversible electric field is required. Moreover, the simpler material model also allows for us to validate our numerical predictions with the experimental results of a micro-sized, multi-layered ferroelectric actuator, in which fewer ferroelectric material parameters are needed, demonstrating the feasibility of the proposed model for realistic modeling applications. The rest of the paper is structured as follows. In Section 2, we will present the background theory where the simple tangent moduli are obtained as a result of the one-to-one simplification. A detailed procedure to obtain the evolution of internal variables at the Gauss-point level, as well as the boundary value problem of ferroelectricity, are presented in Section 3. Based on the FEM formulation, we will demonstrate and validate our numerical results from our in-house code in Section 4 before we make some concluding remarks in Section 5.

## 2. Background Theory

### 2.1. Notations

The continuum model of ferroelectricity will be derived in a Cartesian coordinate system whose basis vectors are $e_{1},e_{2},e_{3}$. Scalar will be denoted by a normal symbol, e.g., a,β, while vector and higher-order tensors are denoted by bold symbols. Conventional tensor calculus notations will be made using Einstein summation, which is used throughout. Inner product over one index (dot product) is denoted by (·), so that $a\cdot b=a_{i}b_{i}\;\left(\forall i,j=1,2,3,\;a\in R^{3},b\in R^{3}\right)$. Inner product over two indices (contraction) is denoted by (:), e.g., $A:b=A_{ij}B_{ij}\;\left(\forall i,j=1,2,3,\;A,B\in R^{3}\times R^{3}\right)$. Dyadic product is denoted by (⊗) so that ${\left(a⊗b\right)}_{ij}=a_{i}b_{j}\;\left(\forall i,j=1,2,3,\;a\in R^{3},b\in R^{3}\right)$.

### 2.2. Basic Equations

Consider a homogeneous domain of ferroelectric material of volume Ω, bounded by the boundary ∂Ω such that $\partial \Omega =\partial \Omega^{u}\cup \partial \Omega^{\varphi }=\partial \Omega^{\sigma }\cup \partial \Omega^{\omega }$ and $\partial \Omega^{u}\cap \partial \Omega^{\sigma }=\emptyset$, $\partial \Omega^{\varphi }\cap \partial \Omega^{\omega }=\emptyset$. $\partial \Omega^{u}$ and $\partial \Omega^{\varphi }$ are the portion of boundary on which displacement **u** and electrical potential φ are prescribed, respectively. Similarly, $\partial \Omega^{\sigma }$ and $\partial \Omega^{\omega }$ are the boundary portion on which mechanical traction **t** and surface charge ω are prescribed, respectively, as shown schematically in Figure 1. It is assumed that the ferroelectric domain is free of body force and charge carriers, so that the governing equations, including the balance of linear momentum and Gauss’s law, are given as follows:
(1a)$$\begin{array}{cc} \nabla \cdot \sigma & =0,\;\mathrm{in}\;\Omega , \end{array}$$
(1b)$$\begin{array}{cc} \nabla \cdot D & =0,\;\mathrm{in}\;\Omega , \end{array}$$
where $\nabla =\frac{\partial }{\partial x_{i}}$ is the Nabla operator, σ is the second-order Cauchy stress tensor, and **D** is the electric displacement vector. The boundary is subjected to the following conditions
(2a)$$\begin{array}{cc} u & =\bar{u},\;\mathrm{on}\;\partial \Omega^{u}, \end{array}$$
(2b)$$\begin{array}{cc} \varphi & =\bar{\varphi },\;\mathrm{on}\;\partial \Omega^{\varphi }, \end{array}$$
(2c)$$\begin{array}{cc} t & =\sigma \cdot n=\bar{t},\;\mathrm{on}\;\partial \Omega^{\sigma }, \end{array}$$
(2d)$$\begin{array}{cc} \omega & =D\cdot n=\bar{\omega },\;\mathrm{on}\;\partial \Omega^{D}, \end{array}$$
in which the $\left(\bar{\circ }\right)$ denotes the prescribed variables; the constitute laws that relate the stress σ and electric displacement **D** to the kinematic variables mechanical strain ε and electric field **E** will be discussed in the next section. Under the assumption of small deformation, the (total) strain is given as the symmetric part of the displacement gradient, as follows
(3)$$\varepsilon =\frac{1}{2}\left(\nabla u+u\nabla \right).$$

The (total) electric field is given as the gradient of scalar electrical potential
(4)E=−∇φ.

In the context of ferroelectricity, these two total kinematic variables can be decomposed into reversible and irreversible parts
(5a)$$\begin{array}{cc} \varepsilon & =\varepsilon^{r}+\varepsilon^{i}, \end{array}$$
(5b)$$\begin{array}{cc} E & =E^{r}+E^{i}, \end{array}$$
where $\varepsilon^{r}$, $E^{r}$ are reversible strain and reversible electric field, respectively, whereas $\varepsilon^{i}$, $E^{i}$ are irreversible strain and irreversible electric field, respectively. The irreversible polarization $P^{i}$ is related to the irreversible electric field $E^{i}$ by
(6)$$P^{i}=-\kappa \cdot E^{i},$$
where κ is the second-order dielectric tensor.

### 2.3. Thermodynamic Consistency

The main ingredient of the proposed formulation is the assumption that the irreversible strain is forced to be in alignment with the irreversible polarization (or indirectly with the irreversible electric field) such that there is a one-to-one correspondence between the irreversible electric field and irreversible strain. As noted in McMeeking [32], this assumption might not be justified in some types of piezoelectric material; however, it can be valid for the majority of piezoelectric materials, including our PZT material, as will be shown in the following sections. Besides, such a one-to-one relation has been utilized by several authors in the phenomenological model [1,22,50]. The one-to-one relation reads as follows: (7)$$\varepsilon^{i}=\frac{\varepsilon^{s}}{2{\left(E^{s}\right)}^{2}}\left(3E^{i}⊗E^{i}-\left(E^{i}\cdot E^{i}\right)I\right),$$
where **I** is the second-order identity tensor, $E^{s}$ and $\varepsilon^{s}$ are the two material properties, namely the saturation electric field and saturation strain, respectively. Equation (Equation 7) resembles the electrostriction effect, where the induced-deformation is volume-preserved.

To phenomenologically describe ferroelectric material, the Helmholtz free energy that takes kinematic variables ε, **E** and internal variables $\varepsilon^{i}$, $E^{i}$ as independent arguments is defined as follows [52]
(8)$$\psi \left(\varepsilon ,\varepsilon^{i},E,E^{i}\right)=\frac{1}{2}\left(\varepsilon -\varepsilon^{i}\right):C:\left(\varepsilon -\varepsilon^{i}\right)+\frac{\beta }{E^{s}}E\cdot e:\left(\varepsilon -\varepsilon^{i}\right)-\frac{1}{2}\left(E-E^{i}\right)\cdot \kappa \cdot \left(E-E^{i}\right)+\bar{\psi }\left(\varepsilon^{i},E^{i}\right),$$
where **C**, **e** and κ are the fourth-order elastic tensor, third-order piezoelectric tensor and second-order dielectric material tensors, respectively. While the elastic and dielectric energy takes the quadratic form (the first and third terms), the piezoelectric coupling energy (the second term) is proportional to the coefficient $\beta =E^{i}\cdot e^{P}$, with $e^{P}$, denoting the normalized direction vector of polarization. This indicates that the electromechanical coupling effect can only take place with non-zero remanent polarization and is fully coupled when the material is fully poled. In Equation (Equation 8), $\bar{\psi }\left(\varepsilon^{i},E^{i}\right)$ defines the hardening potential that will be implicitly defined in the next sections. It should be noted that although the irreversible strain $\varepsilon^{i}$ can be replaced by irreversible electric field $E^{i}$, it will be kept for the sake of clarity.

Within the theory of electro-elastic continuum, the second-law of thermodynamic dictates the following inequality of dissipation
(9)$$D:=\sigma :\dot{\varepsilon }-D\cdot \dot{E}-\dot{\psi }\geq 0,$$
where the $\left(\dot{\circ }\right)$ indicates time derivative. For instance,
(10)$$\dot{\psi }:=\frac{\partial \psi }{\partial \varepsilon }:\dot{\varepsilon }+\frac{\partial \psi }{\partial \varepsilon^{i}}:{\dot{\varepsilon }}^{i}+\frac{\partial \psi }{\partial E}\cdot \dot{E}+\frac{\partial \psi }{\partial E^{i}}\cdot {\dot{E}}^{i}.$$

Under the framework of rational continuum mechanics, upon substituting Equation (Equation 8) into Equation (Equation 9), the constitutive equations for σ and **D** and driving forces $\hat{\sigma }$ and $\hat{D}$ can be obtained
(11a)$$\begin{array}{cc} \sigma :=\frac{\partial \psi }{\partial \varepsilon } & =C:\left(\varepsilon -\varepsilon^{i}\right)+\frac{\beta }{E^{s}}e\cdot E, \end{array}$$
(11b)$$\begin{array}{cc} D:=-\frac{\partial \psi }{\partial E} & =-\frac{\beta }{E^{s}}e:\left(\varepsilon -\varepsilon^{i}\right)+\kappa \cdot \left(E-E^{i}\right), \end{array}$$
(11c)$$\begin{array}{cc} \hat{\sigma }:=-\frac{\partial \psi }{\partial \varepsilon^{i}} & =C:\left(\varepsilon -\varepsilon^{i}\right)+\frac{\beta }{E^{s}}e\cdot E-\frac{\partial \bar{\psi }}{\partial \varepsilon^{i}}, \end{array}$$
(11d)$$\begin{array}{cc} \hat{D}:=-\frac{\partial \psi }{\partial E^{i}} & =-\frac{e^{P}}{E^{s}}E\cdot e:\left(\varepsilon -\varepsilon^{i}\right)-\kappa \cdot \left(E-E^{i}\right)-\frac{\partial \bar{\psi }}{-E^{i}}, \end{array}$$
where the so-called back-stress $\frac{\partial \bar{\psi }}{\partial \varepsilon^{i}}=0$ in our simplified model, $\frac{\partial \bar{\psi }}{\partial E^{i}}=D^{B}$ is referred to as back-electric displacement and will be explicitly detailed in the following sections. In Equations (10) and (11), the dissipation inequality Equation (9) can be reduced to
(12)$$D:=\hat{\sigma }:{\dot{\varepsilon }}^{i}+\hat{D}\cdot {\dot{E}}^{i}\geq 0,$$
which involves the rate of change of the irreversible strain $\varepsilon^{i}$ and electric field $E^{i}$. Therefore, upon utilizing the one-to-one relation from Equation (Equation 7), the reduced dissipation inequality can be further simplified, as follows: (13)$$D=\frac{\varepsilon^{s}}{2{\left(E^{s}\right)}^{2}}\left(6\hat{\sigma }\cdot E^{i}-2\left(I:\hat{\sigma }\right)\right)\cdot {\dot{E}}^{i}+\hat{D}\cdot {\dot{E}}^{i}=\tilde{D}\cdot {\dot{E}}^{i}\geq 0,$$
where
(14)$$\tilde{D}=\underset{D}{\underset{︸}{\frac{\varepsilon^{s}}{2{\left(E^{s}\right)}^{2}}\left(6\hat{\sigma }\cdot E^{i}-2\left(I:\hat{\sigma }\right)\right)}}+\hat{D}=D+\hat{D}$$

To signify when irreversible processes take place, we introduce the dissipation potential
(15)$$\phi :=\frac{{\left(\tilde{D}\cdot e^{P}\right)}^{2}}{{\left(P^{c}\right)}^{2}}-1\leq 0,$$
where $P^{c}$ is the coercive polarization and related to the coercive electric field through the dielectric constant. The evolution of internal variable, i.e., $E^{i}$ thanks to the one-to-one assumption in Equation (Equation 7), occurs in the direction that maximizes the dissipation potential. In other words, by minimizing the following Lagrangian
(16)$$L:=-D\left(\hat{\sigma },\hat{D}\right)+\lambda \phi \left(\hat{\sigma },\hat{D}\right)\to \min,$$
where λ is the Lagrange multiplier, the evolution equation of the irreversible electric field
(17)$${\dot{E}}^{i}=\lambda \frac{\partial \phi }{\partial \tilde{D}}=\lambda \tilde{E},$$
and the loading/unloading conditions
(18)λ≥0,ϕ≤0,λϕ=0,
can be obtained. In Equation (Equation 17), we define
(19)$$\tilde{E}=\frac{\partial \phi }{\partial \tilde{D}}=\frac{2}{{\left(P^{c}\right)}^{2}}\left(\tilde{D}\cdot e^{P}\right)e^{P}$$

To derive tangent moduli from the rate of the constitute laws, using Equations (Equation 17) and (Equation 19), the consistency equation is firstly expressed as
(20)$$\begin{array}{cc} \dot{\phi } & =\frac{\partial \phi }{\partial \tilde{D}}\cdot \dot{\tilde{D}}+\frac{\partial \phi }{\partial E^{i}}\cdot {\dot{E}}^{i}\\ \; & =\tilde{E}\cdot \dot{\tilde{D}}+\lambda \frac{\partial \phi }{\partial E^{i}}\cdot \tilde{E}=0. \end{array}$$

As $\tilde{D}$ is defined as in Equation (Equation 14), the first product of the above equation can be written as
$$\begin{array}{cc} \tilde{E}\cdot \dot{\tilde{D}} & =\tilde{E}\cdot \left(\frac{\partial \tilde{D}}{\partial \varepsilon }\cdot \dot{\varepsilon }+\frac{\partial \tilde{D}}{\partial E}\cdot \dot{E}+\frac{\partial \tilde{D}}{\partial E^{i}}\cdot \dot{E^{i}}\right)\\ \; & =\overset{⏜}{\sigma }:\dot{\varepsilon }+\overset{⏜}{D}\cdot \dot{E}+\lambda \left[\frac{\partial \tilde{D}}{\partial E^{i}}:\left(\tilde{E}⊗\tilde{E}\right)\right], \end{array}$$
where
(21a)$$\begin{array}{cc} \overset{⏜}{\sigma } & =\tilde{E}\cdot \frac{\partial \tilde{D}}{\partial \varepsilon }, \end{array}$$
(21b)$$\begin{array}{cc} \overset{⏜}{D} & =\tilde{E}\cdot \frac{\partial \tilde{D}}{\partial E}. \end{array}$$

Consequently, from the consistency Equation (Equation 20), we can determine the Lagrange multiplier λ, as follows
(22)$$\lambda =\frac{\overset{⏜}{\sigma }:\dot{\varepsilon }+\overset{⏜}{D}\cdot \dot{E}}{\chi },$$
in which the denominator is given by
(23)$$\chi =-[\frac{\partial \tilde{D}}{\partial E^{i}}:\left(\tilde{E}⊗\tilde{E}\right)+\frac{\partial \phi }{\partial E^{i}}]$$

Upon substituting Equation (Equation 23) into Equation (Equation 17), the evolution of the irreversible electric field can be expressed in terms of λ as
(24)$$\begin{array}{c} {\dot{E}}^{i}=\tilde{E}\frac{\overset{⏜}{\sigma }:\dot{\varepsilon }+\overset{⏜}{D}\cdot \dot{E}}{\chi } \end{array}$$

As a result, by taking time derivative of the constitutive laws Equations (11a) and (11b) and making use of (Equation 24), the rate form of constitutive laws can be written as
(25)$$\begin{array}{cc} \dot{\sigma } & =\left[\frac{1}{\chi }\left(\frac{\partial \sigma }{\partial E^{i}}\cdot \tilde{E}\right)⊗\overset{⏜}{\sigma }+C\right]:\dot{\varepsilon }+\left[\frac{1}{\chi }\left(\frac{\partial \sigma }{\partial E^{i}}\cdot \tilde{E}\right)⊗\overset{⏜}{D}+\frac{\beta }{E^{s}}e\right]\cdot \dot{E},\\ \; & =\tilde{C}:\dot{\varepsilon }+{\tilde{e}}^{\sigma }\cdot \dot{E}, \end{array}$$
(26)$$\begin{array}{cc} \dot{D} & =\left[\frac{1}{\chi }\left(\frac{\partial D}{\partial E^{i}}\cdot \tilde{E}\right)⊗\overset{⏜}{\sigma }-\frac{\beta }{E^{s}}e\right]:\dot{\varepsilon }+\left[\frac{1}{\chi }\left(\frac{\partial D}{\partial E^{i}}\cdot \tilde{E}\right)⊗\overset{⏜}{D}+\kappa \right]\cdot \dot{E},\\ \; & ={\tilde{e}}^{D}:\dot{\varepsilon }+\tilde{\kappa }\cdot \dot{E}, \end{array}$$
with
(27a)$$\begin{array}{cc} \tilde{C} & =[\frac{1}{\chi }\left(\frac{\partial \sigma }{\partial E^{i}}\cdot \tilde{E}\right)⊗\overset{⏜}{\sigma }+C], \end{array}$$
(27b)$$\begin{array}{cc} {\tilde{e}}^{\sigma } & =[\frac{1}{\chi }\left(\frac{\partial \sigma }{\partial E^{i}}\cdot \tilde{E}\right)⊗\overset{⏜}{D}+\frac{\beta }{E^{s}}e], \end{array}$$
(27c)$$\begin{array}{cc} {\tilde{e}}^{D} & =[\frac{1}{\chi }\left(\frac{\partial D}{\partial E^{i}}\cdot \tilde{E}\right)⊗\overset{⏜}{\sigma }-\frac{\beta }{E^{s}}e], \end{array}$$
(27d)$$\begin{array}{cc} \tilde{\kappa } & =[\frac{1}{\chi }\left(\frac{\partial D}{\partial E^{i}}\cdot \tilde{E}\right)⊗\overset{⏜}{D}+\kappa ], \end{array}$$
being the tangent elastic, piezoelectric and dielectric tensors. To this end, we established the necessary constitutive laws (in rate-form) that can be used in the boundary value problem described in Section 2.2. However, as can be seen from Equation (27), the tangent moduli depend on the evolution of internal variable $E^{i}$, which is governed by Equations (15), (17) and (18) and will be addressed in the next section. The partial derivatives with respect to $E^{i}$ can be found in Appendix A.

## 3. Finite-Element Formulation

This section presents the finite-element implementation of the proposed model. The key variable that describes hysteresis behavior and determines the tangent moduli, the irreversible electric field $E^{i}$, will be obtained by the return mapping technique. Subsequently, weak-form ferroelectricity and its linearization are introduced and discretized by finite-element approximation. The local (Gauss-point level) and global system of nonlinear equations are solved by the Newton–Raphson method.

### 3.1. Return Mapping

In order to solve the evolution Equation (Equation 17), a backward Euler time marching scheme is employed. Specifically, the time-discretization form of Equation (Equation 17) can be written as
(28a)$$\begin{array}{cc} E_{n+1}^{i} & =E_{n}^{i}+\gamma_{n+1}{\tilde{E}}_{n+1}, \end{array}$$
(28b)$$\begin{array}{cc} \phi_{n+1} & =\frac{{\left({\tilde{D}}_{n+1}\cdot e^{P}\right)}^{2}}{{\left(P^{c}\right)}^{2}}-1, \end{array}$$
(28c)$$\begin{array}{cc} \gamma_{n+1} & \geq 0,\;\phi_{n+1}\leq 0,\;\gamma_{n+1}\phi_{n+1}=0, \end{array}$$
with *n* as the pseudo-time step, $\gamma_{n+1}=\lambda_{n+1}/\Delta t$. Regarding plasticity, the operator splitting technique is employed. Firstly, in the predictor step, the trial state of the irreversible electric field can be obtained by ‘freezing’ the switching process
(29a)$$\begin{array}{cc} \gamma_{n+1}^{\mathrm{trial}} & =0, \end{array}$$
(29b)$$\begin{array}{cc} E_{n+1}^{i,\mathrm{trial}} & =E_{n}^{i}, \end{array}$$
(29c)$$\begin{array}{cc} \varepsilon^{i,\mathrm{trial}} & =\frac{\varepsilon^{s}}{2{\left(E^{s}\right)}^{2}}\left(3E^{i,\mathrm{trial}}⊗E^{i,\mathrm{trial}}-\left(E^{i,\mathrm{trial}}\cdot E^{i,\mathrm{trial}}\right)I\right), \end{array}$$
(29d)$$\begin{array}{cc} {\hat{\sigma }}^{\mathrm{trial}} & =C:\left(\varepsilon -\varepsilon^{i,\mathrm{trial}}\right)+\frac{\beta }{E^{s}}e\cdot E, \end{array}$$
(29e)$$\begin{array}{cc} {\hat{D}}^{\mathrm{trial}} & =-\frac{1}{E^{s}}E\cdot e:\left(\varepsilon -\varepsilon^{i,\mathrm{trial}}\right)-\kappa \left(E-E^{i,\mathrm{trial}}\right), \end{array}$$
(29f)$$\begin{array}{cc} {\tilde{D}}^{\mathrm{trial}} & =\frac{\varepsilon^{s}}{2{\left(E^{s}\right)}^{2}}\left(6{\hat{\sigma }}^{\mathrm{trial}}\cdot E^{i,\mathrm{trial}}-2\left(I:{\hat{\sigma }}^{\mathrm{trial}}\right)\right)+{\hat{D}}^{\mathrm{trial}}, \end{array}$$
(29g)$$\begin{array}{cc} \phi_{n+1}^{\mathrm{trial}} & =\frac{{\left({\tilde{D}}_{n+1}^{\mathrm{trial}}\cdot e^{P}\right)}^{2}}{{\left(P^{c}\right)}^{2}}-1. \end{array}$$

The value of $\phi_{n+1}^{\mathrm{trial}}$ is evaluated, such that when $\phi_{n+1}^{\mathrm{trial}}\leq 0$, the trial states are registered as the true states. Otherwise, if $\phi_{n+1}^{\mathrm{trial}}>0$, the switching process takes place and is governed by the following set of nonlinear equations
(30a)$$\begin{array}{cc} E_{n+1}^{i} & =E_{n}^{i}+\gamma_{n+1}{\tilde{E}}_{n+1}, \end{array}$$
(30b)$$\begin{array}{cc} \phi_{n+1} & =\frac{{\left({\tilde{D}}_{n+1}\cdot e^{P}\right)}^{2}}{{\left(P^{c}\right)}^{2}}-1=0, \end{array}$$
where $x_{n+1}={\left[E_{n+1}^{i},\gamma_{n+1}\right]}^{T}$ are the unknown variables. Let us define a residual vector
(31)$$R_{n+1}=\left[\begin{array}{c} R_{n+1}^{a}\\ R_{n+1}^{b} \end{array}\right]=\left[\begin{array}{c} E_{n+1}^{i}-\left(E_{n}^{i}+\gamma_{n+1}{\tilde{E}}_{n+1}\right)\\ \frac{{\left({\tilde{D}}_{n+1}\cdot e^{P}\right)}^{2}}{{\left(P^{c}\right)}^{2}}-1 \end{array}\right].$$

Then, the solution of the system of nonlinear equations Equation (30) can be updated in the local (Gauss-point level) Newton–Raphson procedure as follows
(32)$$x_{n+1}^{k+1}=x_{n+1}^{k}-{\left[\frac{\partial R_{n+1}}{\partial x}\right]}^{-1}\cdot R\left(x_{n+1}^{k}\right),$$
with *k* as the Newton iteration step and
(33)$$\frac{\partial R_{n+1}}{\partial x}=\left[\begin{array}{cc} \frac{\partial R^{a}}{\partial E_{n+1}^{i}} & \frac{\partial R^{a}}{\partial \gamma_{n+1}}\\ \frac{\partial R^{b}}{\partial E_{n+1}^{i}} & \frac{\partial R^{b}}{\partial \gamma_{n+1}} \end{array}\right]=\left[\begin{array}{cc} I-\gamma_{n+1}\frac{\partial {\tilde{E}}_{n+1}}{\partial E_{n+1}^{i}} & -{\tilde{E}}_{n+1}\\ \frac{2}{{\left(P^{c}\right)}^{2}}{\tilde{D}}_{n+1}\cdot \frac{\partial {\tilde{D}}_{n+1}}{\partial E_{n+1}^{i}} & 0 \end{array}\right]$$

For the sake of clarity, the return mapping algorithm is presented in Algorithm 1. Once the evolution of the irreversible electric field is determined, the obtained solution $E_{n+1}^{i}$ is then used to calculate the tangent moduli according to Equation (27), which is subsequently utilized to compute the tangent matrices, as will be shown in the next section.
**Algorithm 1:** Return mapping algorithm.
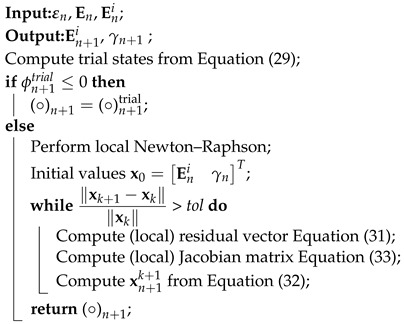


### 3.2. Weak-Form and Solving Procedure

With the definition of σ and **D**, the weak form of ferroelectrostatic can be written as
(34)$$\begin{array}{c} \delta \pi =\delta \psi -\delta \psi^{\mathrm{ext}}=\int_{\Omega }\sigma :\delta \varepsilon dV-\int_{\Omega }D\cdot \delta EdV-\int_{\partial \Omega }\delta u\cdot \bar{t}dA+\int_{\partial \Omega }\delta \varphi \bar{\omega }dA, \end{array}$$
in which $\pi =\psi -\psi^{\mathrm{ext}}$ is defined as the total potential.

Linearizing the weak form, we can obtain
(35)$$\begin{array}{cc} \Delta \delta \pi & =\int_{\Omega }\delta \varepsilon :\frac{\partial \sigma }{\partial \varepsilon }:\Delta \varepsilon dV+\int_{\Omega }\delta \varepsilon :\frac{\partial \sigma }{\partial E}:\Delta EdV-\int_{\Omega }\delta E\cdot \frac{\partial D}{\partial \varepsilon }:\Delta \varepsilon dV-\int_{\Omega }\delta E\cdot \frac{\partial^{2}D}{\partial E\partial \cdot \Delta E}dV \end{array}$$

To perform finite-element analysis, the primary variables **u**, φ and their variations δu, δφ are approximated by linear shape functions (note that hereafter, the matrix form is adapted)
(36a)$$\begin{array}{cc} u & =N^{u}\tilde{u},\;\delta u=N^{u}\delta \tilde{u}, \end{array}$$
(36b)$$\begin{array}{cc} \varphi & =N^{\varphi }\tilde{\varphi },\;\delta \varphi =N^{\varphi }\delta \tilde{\varphi }, \end{array}$$
so that, from the kinematics relation, the strain and electric field can be discretized as follows
(37a)$$\begin{array}{cc} \varepsilon & =B^{u}\tilde{u},\;\delta \varepsilon =B^{u}\delta \tilde{u}, \end{array}$$
(37b)$$\begin{array}{cc} E & =B^{\varphi }\tilde{\varphi },\;\delta E=B^{\varphi }\delta \tilde{\varphi }, \end{array}$$
where the shape functions matrices $N^{u}$, $N^{\varphi }$ and differential shape functions matrices $B^{u}$, $B^{\varphi }$ can be found in the Appendix B for more details.

Using the approximations and employing the arbitrariness of the test functions, the residual vector $G\left(u,\varphi \right)={\left[G^{u},G^{\varphi }\right]}^{T}$ can be obtained as
(38a)$$\begin{array}{cc} G^{u} & :=G^{u,\mathrm{int}}-G^{u,\mathrm{ext}}=\int_{\Omega }B^{u,T}\sigma \left(u,\varphi \right)dV-\int_{\partial \Omega }N^{u,T}\bar{t}dA=0, \end{array}$$
(38b)$$\begin{array}{cc} G^{\varphi } & :=G^{\varphi ,\mathrm{int}}-G^{\varphi ,\mathrm{ext}}=-\int_{\Omega }B^{\varphi ,T}D\left(u,\varphi \right)dV+\int_{\partial \Omega }N^{\varphi ,T}\bar{\omega }dA=0, \end{array}$$
where the *T* superscript indicates the transpose matrix. The nonlinear equation is solved by the Newton–Raphson approach, in which the incremental nodal degrees of freedom can be obtained from
(39)$$\left[\begin{array}{cc} K^{uu} & K^{u\varphi }\\ K^{\varphi u} & K^{\varphi \varphi } \end{array}\right]\left[\begin{array}{c} \Delta \tilde{u}\\ \Delta \tilde{\varphi } \end{array}\right]=\left[\begin{array}{c} G^{u,\mathrm{ext}}\\ G^{\varphi ,\mathrm{ext}} \end{array}\right]-\left[\begin{array}{c} G^{u,\mathrm{int}}\left({\tilde{u}}_{n+1}^{l},{\tilde{\varphi }}_{n+1}^{l}\right)\\ G^{\varphi ,\mathrm{int}}\left({\tilde{u}}_{n+1}^{l},{\tilde{\varphi }}_{n+1}^{l}\right) \end{array}\right],$$
where *l* is the (global) Newton–Raphson iteration and, in view of Equation (Equation 35), the tangent stiffness matrices are given as
(40a)$$\begin{array}{cc} K^{uu} & =\int_{\Omega }B^{u,T}\tilde{C}B^{u}dV, \end{array}$$
(40b)$$\begin{array}{cc} K^{u\varphi } & =-\int_{\Omega }B^{u,T}{\tilde{e}}^{\sigma }B^{\varphi }dV, \end{array}$$
(40c)$$\begin{array}{cc} K^{\varphi u} & =-\int_{\Omega }B^{\varphi ,T}{\tilde{e}}^{D}B^{u}dV, \end{array}$$
(40d)$$\begin{array}{cc} K^{\varphi \varphi } & =-\int_{\Omega }B^{\varphi ,T}\tilde{\kappa }B^{\varphi }dV, \end{array}$$
where $\tilde{C}$, ${\tilde{e}}^{\sigma }$, ${\tilde{e}}^{D}$ and $\tilde{\kappa }$ are the tangent elastic, piezoelectric and dielectric moduli, as defined in Equation (Equation 25). Details of matrix forms and necessary partial derivative terms can be found in the  Appendix B. The overall finite element implementation is summarized in Algorithm 2.
**Algorithm 2:** Finite element framework of ferroelectricty.
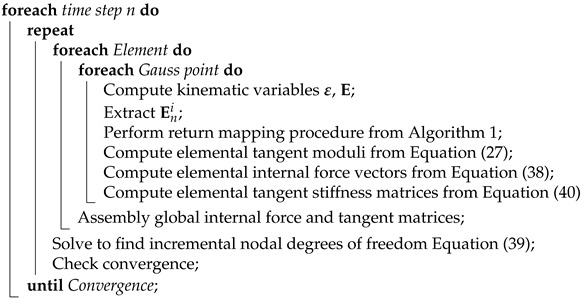


## 4. Numerical Examples

In order to illustrate and validate the ferroelectric responses predicted by our model, various numerical examples will be performed and compared with other numerical and experimental results. We will first demonstrate the hysteresis and mechanical depolarization behaviors in a 1D setting under stress-free or electric-field-free assumption. We will further demonstrate and compare both hysteresis and mechanical depolarization with an experimental measurement of a 3D ferroelectric cube. Finally, the feasibility of our proposed model will be validated to the model the micro-ferroelectric actuator.

### 4.1. Analytical Solutions

Before performing finite-element analysis, let us examine the nonlinear response of ferroelectricity from a simple 1D setting, which might be useful to calibrate hardening parameters. To illustrate the hysteresis behavior, a set of material parameters of a typical ferroelectric ceramic are chosen and shown in Table 1. Note that the coercive field can be found from $E^{c}=P^{c}/\kappa$ or, inversely, the saturation polarization can be derived from $P^{s}=E^{s}\kappa$. Similar to [55], the back-electric displacement $D^{B}$ is chosen as
(41)$$D^{B}=\frac{\partial \bar{\psi }}{\partial E^{i}}=kE^{i}+a\mathrm{atanh}\left(\frac{E^{i}}{bE^{s}}\right),$$
with material parameters k,a,b given in Table 1, so that $\tilde{D}$ takes the form
(42)$$\tilde{D}=\frac{\varepsilon^{s}}{{\left(E^{s}\right)}^{2}}\hat{\sigma }E^{i}-\frac{E}{E^{s}}e\left(\varepsilon -\varepsilon^{i}\right)-\kappa \left(E-E^{i}\right)-kE^{i}-a\mathrm{atanh}\left(\frac{E^{i}}{bE^{s}}\right)$$

By assuming a stress-free state, one can obtain the reversible strain from the constitutive relation
(43)$$\varepsilon^{r}=\varepsilon -\varepsilon^{i}=\frac{-E^{i}}{E^{s}}\frac{e}{C}E$$

Consequently, the hysteresis loops can be described by solving for $E^{i}$ from the combination of Equation (Equation 42) and the consistency equation Equation (Equation 15), i.e., $\tilde{D}=\pm P_{c}$, as a function of applied electric field *E*, whose profile is shown in Figure 2a, with $E^{max}=2E^{c}$
(44)$$\frac{e^{2}}{C{\left(E^{s}\right)}^{2}}E^{2}E^{i}-\kappa \left(E-E^{i}\right)-kE^{i}-a\mathrm{atanh}\left(\frac{E^{i}}{bE^{s}}\right)=\pm P_{c}$$

The solution was obtained numerically using function fzero in $Matlab^{©}$. The evolution of $E^{i}$, accompanied by the yield function along the applied electric field, is shown in Figure 2a. Initially, at the state ➀, the material is unpoled, meaning that there is zero net spontaneous polarization and zero remanent electric field (or remanent polarization). The value of the yield function at the initial state is −1. As the external electric field increases from ➀ to ➁, the yield function increases from −1 to 0 and the material still behaves linearly. Note that piezoelectric coupling does not exist in this range, as the internal variables do not develop. When the electric field reaches the coercive value at state ➁, a switching process starts to take place, in which the yield function reaches 0 value and the irreversible electric field starts to nonlinearly develop from 0 to $-E^{s}$ from ➁ to ➂. When the electric field decreases from its maximum value at ➂ to zero at ➃, the yield function becomes negative and the material shows a linear piezoelectric effect as the irreversible electric field is ’locked’ at $-E^{s}$, meaning that the material undergoes upward-aligned polarization. As the electric field further decreases in the opposite direction from state ➃, the yield function reaches zero, signifying that a switching phenomenon takes place. Between states ➃ and ➄, the irreversible electric field $E^{i}$ evolves nonlinearly and changes its orientation from $-E^{s}$ to zero at the coercive threshold (➄ is the opposite of ➁). When the electric field decreases to the minimum value at state ➅, the yield function remains at 0 and the irreversible electric field continues to nonlinearly increase until it reaches the value $E^{s}$, where the polarization is completely switched to the opposite direction. This downward-aligned polarization remains unchanged when the electric field increases again, from the minimum at ➅ to zero at ➆, and the material will again behave as a linear piezoelectric material but with the opposite poling direction from that of states ➂–➃. Apparently, to switch the poling direction, an electric field must be applied in the opposite direction, as can be seen again at states ➁–➂.

In the current setting, all other variables can be described upon the obtained irreversible electric field $E^{i}$, including the hysteresis loops of polarization in Figure 2b and mechanical strain in Figure 2c. Specifically, the one-to-one relation (Equation 7) is used to calculate the irreversible strain $\varepsilon^{i}$, which is subsequently utilized in Equation (Equation 43) to compute reversible and total strains, $\varepsilon^{r}$ and ε, respectively. Meanwhile, the irreversible polarization $P^{i}$ and electric displacement *D* can be computed from Equations (Equation 6) and ([Disp-formula FD11b-micromachines-14-01355]), which, in a 1D setting, can be written as
(45a)$$\begin{array}{cc} P^{i} & =-\kappa E^{i}, \end{array}$$
(45b)$$\begin{array}{cc} D & =-\frac{\beta }{E^{s}}e\left(\varepsilon -\varepsilon^{i}\right)+\kappa \left(E-E^{i}\right) \end{array}$$

Next, we demonstrate the mechanical depolarization characteristics. In this case, the initial state of the ferroelectric material is assumed to be the poled state, corresponding to ➃ in Figure 2a. In the poled state, a compressive (negative) stress, whose profile in shown in Figure 3a with $\sigma_{max}=5\sigma_{c}$, where $\sigma_{c}$ is the coercive stress, is applied, and the remanent electric field is determined from the following nonlinear consistency equation
(46)$$\frac{\varepsilon^{s}}{{\left(E^{s}\right)}^{2}}\hat{\sigma }E^{i}+\kappa E^{i}-kE^{i}-a\mathrm{atanh}\left(\frac{E^{i}}{bE^{s}}\right)=\pm P^{c}.$$

Similarly, a built-in fzero Matlab function was used to solve $E^{i}$ under the applied σ. The change in $E^{i}$, together with the change in the yield function, are shown in Figure 3a. When the compressive stress σ is applied, the switching process takes place as the yield function remain at zero, i.e., ϕ=0, and in the manner that the negative irreversible electric field $E^{i}$ is increased, corresponding to the reduction in positive electric polarization. In the beginning, where the compressive stress is still small, $E^{i}$ only changes slightly, as it is in the ’locked’ state. Afterwards, it changes rapidly to a fixed value as the magnitude of compressive stress reaches $5\sigma_{c}$. Upon the decrease in compressive stress, the yield function becomes negative ϕ<0 and material should behave linearly with a remanent electric field of 28% as its maximum value. The non-zero remanent electric field that occurs as the stress returns to zero is a deficiency of the proposed model, as explained by Equation (Equation 46). This drawback is similar to what has been observed in [32], where only the positive remanent strain can be aligned with the remanent polarization. As a result, the proposed model is limited to describing the mechanical depolarization induced by large compressive stress, as will be shown in the following examples.

Nevertheless, from the evolution result of $E^{i}$, we can determine the electric displacement and total strain as follows
(47a)$$\begin{array}{cc} \varepsilon & =\left(\varepsilon -\varepsilon^{i}\right)+\varepsilon^{i}=\frac{\sigma }{C}+\beta {\left(E^{i}\right)}^{2}, \end{array}$$
(47b)$$\begin{array}{cc} D & =-\frac{\beta }{E^{s}}e\frac{\sigma }{C}+\kappa \left(E-E^{i}\right), \end{array}$$
where the reversible strain at the zero-electric field is obtained from the constitutive law ([Disp-formula FD11a-micromachines-14-01355]) for 1D settings as $\left(\varepsilon -\varepsilon^{i}\right)=\frac{\sigma }{C}$. The results characterizing mechanical depolarization are shown in Figure 3b,c. We note that the remaining 28% irreversible electric field corresponds to the remaining 0.08357 C/m^2^ electric displacement and 0.0155 mechanical strain.

### 4.2. Ferroelectric Cube

In this section, we illustrate and compare our simplified model with the fully coupled model from [52] and experimental measurement. The schematic of the example is shown in Figure 4, where the bottom surface of the cube with size L=10 mm is mechanically constrained and electrically grounded; meanwhile, the top surface is subjected to electric potential or compression force. The cube is assumed to be made from lead lanthanum zirconate titanate (PLZT), whose material parameters are adapted from [52] as an 3D extension of the 1D experimental setup [5] and shown in Table 2. In addition, the hardening function is also extended from 1D (Equation (Equation 41)) to 3D, such that the back-electric displacement is given as
(48)$$D^{B}=\frac{\partial \bar{\psi }}{\partial E^{i}}=\left(k\beta +a\mathrm{atanh}\left(\frac{\beta }{bE^{s}}\right)\right)e^{P},$$
where β is given in Equation (Equation 8) and $e^{P}={\left[0,0,1\right]}^{T}$. The hardening function in Equation (Equation 48) will be used for the rest of the paper.

Similar to the previous 1D ferroelectric bar example, hysteresis behaviors and mechanical depolarization are demonstrated. In the first loading case, an electric field that has similar profile to Figure 2a but with maximum value of $8\times 10^{5}$ V/m is applied, which is equivalent to a electrical potential of 8 V, as prescribed on the top electrode. The applied electric field results in hysteresis behaviors of polarization and mechanical strain, as depicted in Figure 5a,b, showing the excellent agreement between the results from our simplified model and both the fully coupled model and experimental measurement.

To study the mechanical depolarization in our model, the initial state is assumed to be the poled state of the first loading case, i.e., initial $E_{3}^{i}=-E^{s}$. With respect to this initial state, the total electric displacement and strain are equal to their respective saturation values. Upon this state, a compression force whose magnitude increases from 0 to 2125N, then decreases back to 0, is applied, which is similar to the compressive stress profile in Figure 3a. The largest compressive force value induces a compressive stress that is double the coercive stress. Under this compressive stress state, the sample is depolarized, in which electric displacement and total strain decrease with the increase in compressive force before approaching nearly zero values under the unloading process. Depolarization behaviors are illustrated in Figure 5c,d and are, again, compared with the fully coupled model [52] and experimental measurements [5]. While the proposed method can yield acceptable depolarization results for small compressive stresses, its deficiency, as explained in the previous section, emerges at large compressive stresses, such that larger remanent polarization and strain are predicted as compared to the experiment and fully coupled model.

### 4.3. Ferroelectric Actuator

In this example, we demonstrate the feasibility of using our proposed model in the simulation of a micro-actuator that consists of a sputtering PZT layer of 2 µm thickness on top of a silicon substrate of thickness 725 µm that is loaded into a clamping device, as shown in Figure 6. The test structure is essentially a cantilever beam of length of length and width 21mm×5mm. The PZT layer is sandwiched between two 100 nm Platinum (Pt) electrodes, deposited by the physical vapor deposition (PVD) technique. Since the electrode thickness is much smaller than the PZT layer and the substrate, their mechanical contribution can be ignored in the simulation. While the bottom Pt electrode covers the bottom PZT layer, the top Pt electrode only partially covers the area 14mm×4mm of the top surface of the PZT layer. The actuator is driven by an AC electric field through the thickness direction of the PZT layer by grounding the bottom electrode and applying alternating voltages (at 10 Hz) to the top electrode, which is much lower than the resonance frequency of our device (about 1.6 kHz); hence, this is appropriate for the quasi-static regime of our model. By measuring the induced out-of-plane displacement from applied voltage using Laser Doppler Vibrometer (Polytec LDV), the performance of the micro-actuator can be assessed.

We modelled the micro-actuator with the hexagonal element mesh of the micro-actuator, as shown in Figure 6c. The left surface of the beam is fixed, whereas the electric potential of bottom nodes of PZT layer are set to be zero while the top nodes with x<14mm are prescribed with non-zero voltage to represent the partial top electrode coverage. As the driving frequency is low, the model is assumed to be quasi-static. In our model, the material properties of the stack are given in Table 3, where only the active PZT layer has a nonlinear ferroelectric response and the other layers are assumed to be linearly elastic with 2 Lamé parameters. Note that the ferroelectric parameters of PZT layer are chosen to match with measured hysteresis loops. Under the applied voltage of 40V across the PZT layer thickness or equivalently electric field $E_{3}=20\times 10^{6}$ V/m, we obtain the polarization hysteresis and displacement butterfly loops from the experiment and numerical model shown in Figure 7. The numerical model can make good predictions compared to the measured data. However, while the mathematical description yields a symmetric hysteresis loop with respect to both electric field and polarization axes, i.e., the positive and negative remanent polarization has the same magnitude. The symmetric hysteresis profiles are also often observed in an experimental setting for ’bulk’ piezoelectric ceramic. However, this is not always the case for a realistic thin piezoelectric film, where the asymmetric hysteresis profile could be caused by different underlying physical effects, such as the existence of a dead layer or trapped charges (interested readers can find the description of a deformed hysteresis loop in ferroelectric material in [56,57]). Such charge imperfections are likely to give rise to an additional built-in electric field, which is different to the internal irreversible electric field or the applied external electric field in the scope of our work. It should also be noted that the phase-field approach would be more suitable to describe these microscopic events [58,59]; however, it is also possible to include, for instance, a space charge carrier in the phenomenological model [60].

## 5. Conclusions

In this paper, we presented a theoretical framework and numerical implementation of a simplified phenomenological macroscopic model of a ferroelectric actuator. The key ingredient of the proposed model is based on the alignment of the strain with the orientation of the polarization such that a simple one-to-one correspondence between the irreversible strain and irreversible electric field is assumed, which facilitates not only both theoretical derivation of internal variable evolution and consistent tangent moduli for the implementation aspect but also the number of fitting parameters. We have demonstrated several numerical examples, including analytical expression in some cases, and one-dimensional and three-dimensional problems, to evaluate and validate our proposed model with the fully coupled model and experimental measurement. While our model is in good agreement with the results for bulk piezoelectric material response as compared the previously reported results, it suffers similar difficulties in capturing the mechanical depolarization at large compressive stress. Additionally, the proposed model does not resolve ferroelastic behavior under the assumption that mechanical stress only induces reversible polarization, rather than dipole-switching. Therefore, the current model is suitable for hard piezoelectric materials or soft compounds under moderate stress levels. Furthermore, as the main goal of the paper is to demonstrate the feasibility of modeling a micro-actuator, the current model can only produce symmetric hysteresis loops. However, the measured hysteresis loops can be asymmetric due to the different underlying physical effects, such as the existence of trapped charge or free-charge migration in asymmetric stack configurations. Thus, in future research, we plan to extend our current work to incorporate such effects to devise a more suitable numerical tool for the development of thin-film ferroelectric actuators.

## Figures and Tables

**Figure 1 micromachines-14-01355-f001:**
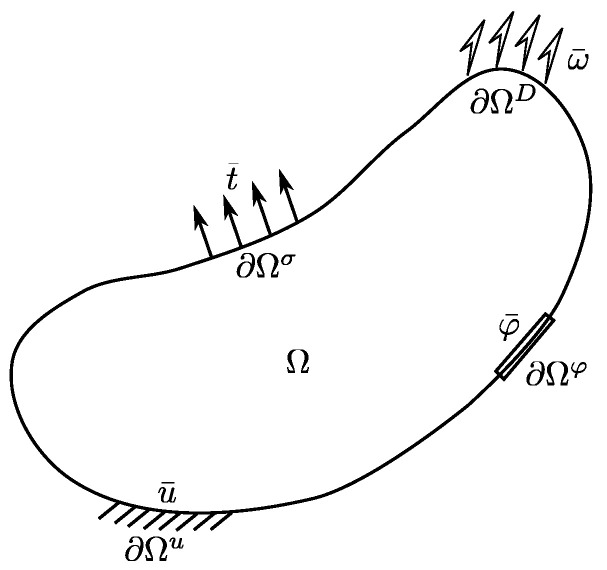
Schematic of a continuum ferroelectric body.

**Figure 2 micromachines-14-01355-f002:**
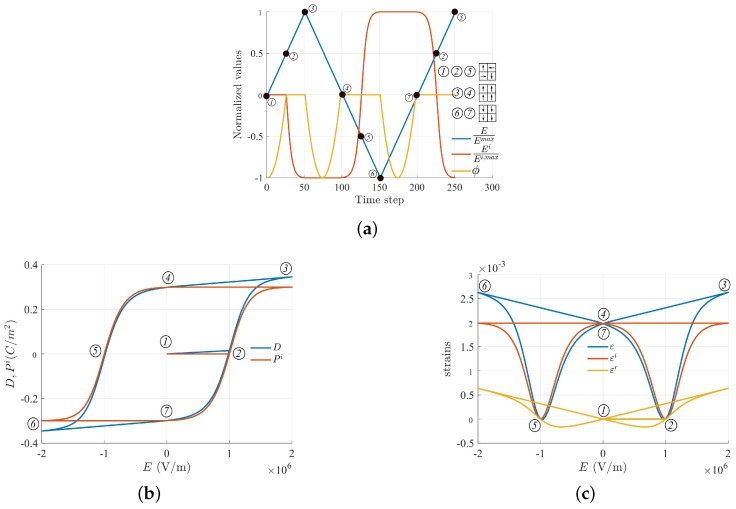
(**a**) Evolution of yield function ϕ and internal variable $E^{i}$ in stress-free conditions, which are electric field-driven with the profile shown as a blue curve. The black arrow schematically represents the orientation of the polarization of microscopic unit-cells, which attribute to the overall polarized states ➀, ➁, etc. (**b**,**c**) Polarization hysteresis and strain butterfly loops were obtained from the solution of Equation (Equation 44).

**Figure 3 micromachines-14-01355-f003:**
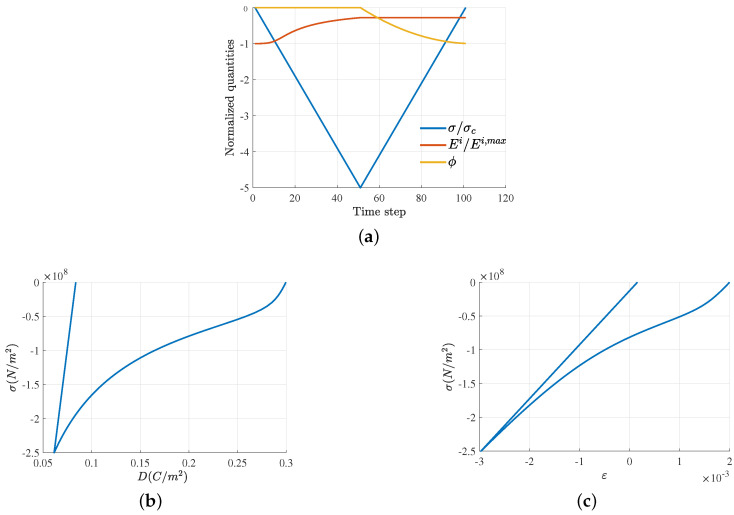
(**a**) Evolution of yield function ϕ and internal variable $E^{i}$ in a zero-electric field, with compressive stress with profile shown as a blue curve. (**b**,**c**) The mechanical depolarization of electric displacement and mechanical strain obtained from the solution of Equation (Equation 46).

**Figure 4 micromachines-14-01355-f004:**
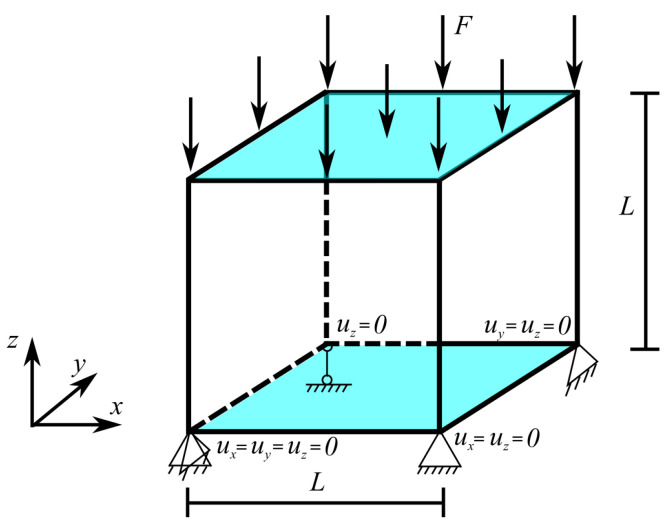
Schematic of a PLZT cube under mechanical constraint in the bottom and applied voltage difference between the bottom and top electrodes.

**Figure 5 micromachines-14-01355-f005:**
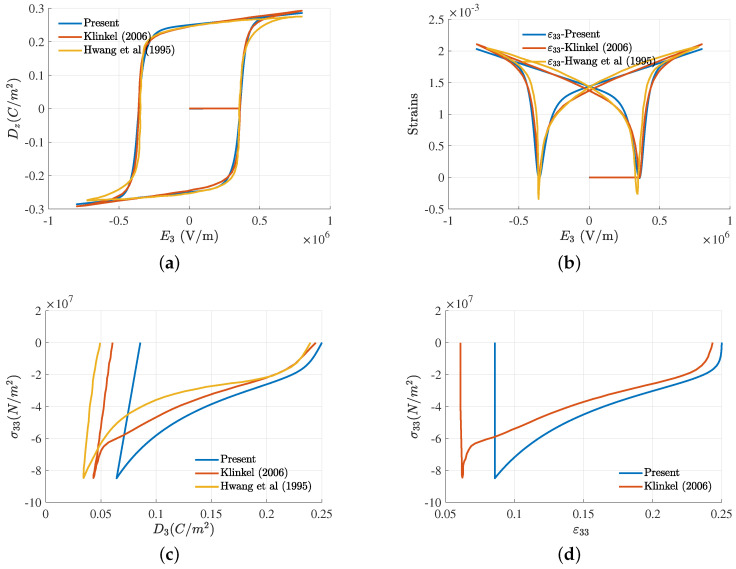
Comparison of ferroelectric and mechanical depolarization response of the problem described in Figure 4, with results from the fully coupled model from Klinkel [52] and experimental measurement from Hwang et al. [5]. (**a**,**b**) Hysteresis behavior; (**c**,**d**) mechanical depolarization behavior of fully poled material under compressive stress $\sigma_{33}$ of electric displacement and strain components, $D_{3}$ and $\varepsilon_{33}$.

**Figure 6 micromachines-14-01355-f006:**
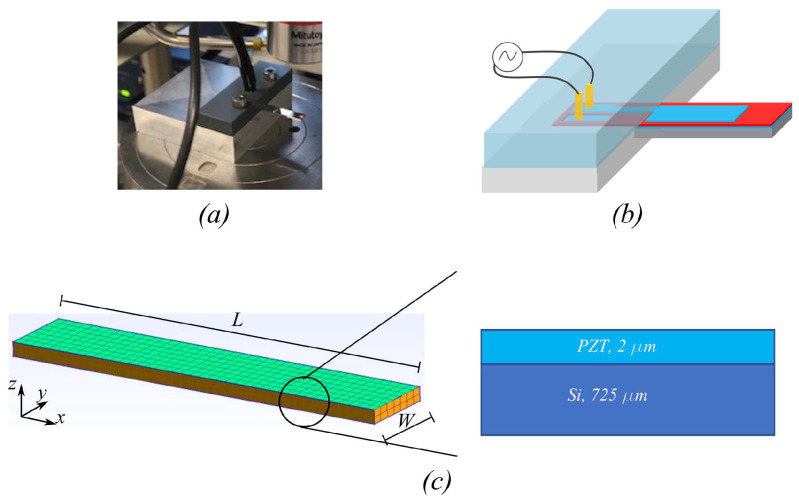
Modeling of ferroelectric micro-actuator. (**a**) Clamped micro-actuator sample. (**b**) Schematic of the micro-actuator. (**c**) Mesh and stack configuration of the micro-actuator.

**Figure 7 micromachines-14-01355-f007:**
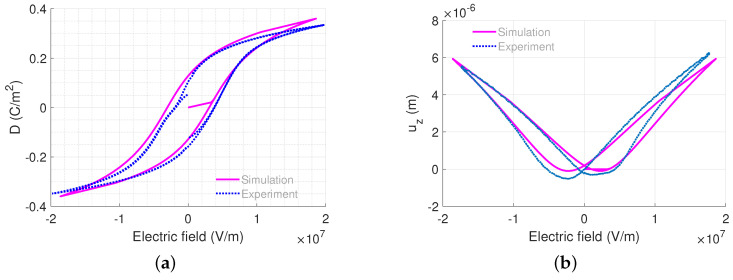
Comparison of the ferroelectric response of the micro-actuator depicted in Figure 6 between simulation and measurement. (**a**) Polarization hysteresis loop. (**b**) Butterfly loop of displacement at the tip of the top electrode.

**Table 1 micromachines-14-01355-t001:** Material parameters of PZT ceramic [52].

$C(N/m^{2})$	$e(C/m^{2})$	$\kappa (C^{2}/\left({\mathrm{Nm}}^{2}\right))$	$P^{c}\left(C/m^{2}\right)$	$E^{s}\left(V/m\right)$	$k(C^{2}/\left({\mathrm{Nm}}^{2}\right))$	*b*	$a(C^{2}/\left({\mathrm{Nm}}^{2}\right))$	$\sigma_{c}\left(N/m^{2}\right)$
$80\times 10^{9}$	25.5802	$15\times 10^{-9}$	$1.5\times 10^{-2}$	$2\times 10^{7}$	$14.985\times 10^{-9}$	1.001	0.053	$50\times 10^{6}$

**Table 2 micromachines-14-01355-t002:** Material properties of PLZT for the problem depicted in Figure 4. Note that the coefficients are given in terms of Voigt notation for transversely isotropic material.

Elastic moduli $10^{9}\;\left(N/m^{2}\right)$					Piezoelectric constants $\left(C/m^{2}\right)$			
$C_{11}$	$C_{12}$	$C_{13}$	$C_{33}$	$C_{44}$	$e_{13}$	$e_{33}$	$e_{15}$	
109	59	59	109	109	−14.96	50.116	38.148	
Dielectric permittivity			Ferroelectric parameters					
$\left(10^{-9}\;C/\mathrm{Vm}\right)$								
$\kappa_{11}$	$\kappa_{33}$		$P^{s}\;\left(C/m^{2}\right)$	$E^{c}\;\left(C/m^{2}\right)$	$\varepsilon^{s}$	k(C/Vm)	*b*	a(C/Vm)
1.125	1.125		0.25	$3.6\times 10^{5}$	0.00144	$1.1249\times 10^{-9}$	1.0001	$1.0097\times 10^{-4}$

**Table 3 micromachines-14-01355-t003:** Material properties of the multilayer micro-actuator depicted in Figure 6.

PZT Layer								
Elastic moduli $\left(10^{9}\;N/m^{2}\right)$					Piezoelectric constants $\left(C/m^{2}\right)$			
$C_{11}$	$C_{12}$	$C_{13}$	$C_{33}$	$C_{44}$	$e_{31}$	$e_{33}$	$e_{15}$	
127.2	80.2	84.7	117.4	23	−16.0	11.7	17	
Dielectric permittivity			Ferroelectric parameters					
$\left(10^{-9}\;C/\mathrm{Vm}\right)$								
$\kappa_{11}$	$\kappa_{22}$		$P^{s}\;\left(C/m^{2}\right)$	$E^{c}\;\left(C/m^{2}\right)$	$\varepsilon^{s}$	$k\;(10^{-9}C/\mathrm{Vm})$	*b*	a(C/Vm)
15.1	6.2		0.36	$4.8\times 10^{6}$	0	6.2	1/tan(0.3π)	1.1 $P^{c}$
Silicon <110> direction								
Elastic moduli $\left(10^{9}\;N/m^{2}\right)$								
$C_{11}$	$C_{12}$	$C_{13}$	$C_{33}$	$C_{44}$	$C_{66}$			
194.5	35.7	64.1	165.7	79.6	50.9			

## Data Availability

The data supporting the reported results are available on request from the corresponding author B.H. Nguyen.

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
