# Peer review of "Simplified Phenomenological Model for Ferroelectric Micro-Actuator"

_micromachines, 2023, doi:10.3390/mi14071355_

Round 1

Reviewer 1 Report

1、What‘s the advantages of the phenomenological model presented in this paper with other techniques.The simulation result need to be compared with current techniques.

2、The innovation of this the phenomenological model  must be  extracted.

3、MEMS technology is the microscale sensor or actuator. However, the phenomenological model presented in this paper is established for the macroscopic model of a nonlinear ferroelectric actuator.  Is there any confilcs?

Author Response

We thank for the Reviewer's comments. Please see section 1 of the attachment for our responses. 

Reviewer 2 Report

The authors developed a simplified model to describe the ferroelectric actuators. Compared with the previous fully-coupled model, this approach is simpler and can correspond well with experimental results. This model proved to be effective at low frequency electrical driven mode, it is with highly practical significance. I recommend accepting this study in Micromachins special issue, but the authors need some more illustration and clarification for a few issues as follows.

1. In the experimental part of figures, such as Figure 7, the unit given on X-axis is V, while in the previous parts, it is V/m. It would be better for the authors to unify the units to avoid confusion.

2. The authors are suggested to give more details in the sample fabrication part, including how the electrodes are deposited onto the material surface, the thickness of the electrode layer, and if the PZT/Si interface affects the mechanical properties of the actuator.

3. The actuation frequency of the given example is 10Hz, which is relatively low and limits the application of the model, the authors are hoped to discuss the model performance at higher frequencies.

Author Response

We thank for the Reviewer's comment. Please see section 2 of the attachment for our responses.

Reviewer 3 Report

In the manuscript of micromachines-2460767, the authors propose a simplified phenomenological macroscopic model of a nonlinear ferroelectric actuator, which is based on a direct relation between the irreversible strain and irreversible electric field, consequently irreversible polarization. The proposed model is then implemented into a finite element framework, in which the main features such as local return mapping and the tangent moduli, are derived. The outcomes of the model are compared and validated with experimental data. I do believe that the development presented in this paper can become a useful tool for the modeling of nonlinear ferroelectric actuators. The research is well organized, and the manuscript is well written. The following comments are provided to the authors for minor revision.

1. In section of introduction, the novelty of this research should be clarified deeply.

2. In the section of discussion, the deficiencies and shortcomings of this macroscopic model should be objectively stated.

3. The style of reference should be checked carefully.

Some typos:

1. In Fig6c, PZT(2 um) and Si (725 um) should be rewrited as PZT(2 μm) and Si (725 μm).

Author Response

We thank for the Reviewer's comments. Please see section 3 of the attachment for our responses.
